# High-performance nanomaterials formed by rigid yet extensible cyclic β-peptide polymers

Kenan P. Fears [1], Manoj K. Kolel-Veetil[1], Daniel E. Barlow[1], Noam Bernstein[2], Christopher R. So[1], Kathryn J. Wahl[1], Xianfeng Li[3], John L. Kulp III[1,4], Robert A. Latour[3] & Thomas D. Clark[1]

Organisms have evolved biomaterials with an extraordinary convergence of high mechanical strength, toughness, and elasticity. In contrast, synthetic materials excel in stiffness or extensibility, and a combination of the two is necessary to exceed the performance of natural biomaterials. We bridge this materials property gap through the side-chain-to-side-chain polymerization of cyclic β-peptide rings. Due to their strong dipole moments, the rings self-assemble into rigid nanorods, stabilized by hydrogen bonds. Displayed amines serve as functionalization sites, or, if protonated, force the polymer to adopt an unfolded conformation. This molecular design enhances the processability and extensibility of the biopolymer. Molecular dynamics simulations predict stick-slip deformations dissipate energy at large strains, thereby, yielding toughness values greater than natural silks. Moreover, the synthesis route can be adapted to alter the dimensions and displayed chemistries of nanomaterials with mechanical properties that rival nature.

[1] Chemistry Division, U.S. Naval Research Laboratory, 4555 Overlook Ave SW, Washington, DC 20375, USA. [2] Materials Science & Technology Division, U.S. Naval Research Laboratory, 4555 Overlook Ave SW, Washington, DC 20375, USA. [3] Department of Bioengineering, Clemson University, 301 Rhodes Research Center, Clemson, SC 29634, USA. [4] Present address: Baruch S. Blumberg Institute, 3805 Old Easton Road, Doylestown, PA 18902, USA. Correspondence and requests for materials should be addressed to K.P.F. (email: kenan.fears@nrl.navy.mil) or to M.K.K-V. (email: manoj.kolel-veetil@nrl.navy.mil)

Nanostructured biopolymers, formed by the linear aggregation of proteins, provide mechanical strength and chemical functionality for a myriad of intracellular and extracellular purposes[1–5]. Generally, natural biopolymers fall into one of two categories in terms of their mechanical properties: (1) high elastic modulus ($E_{init} > 1$ GPa) and low extensibility ($\varepsilon_{max} < 20\%$), or (2) low modulus ($E_{init} < 0.5$ GPa) and high extensibility ($\varepsilon_{max} > 100\%$)[2]. While there are industrial materials that, by far, exceed these values for stiffness (e.g., single-walled carbon nanotubes[6]) or extensibility (e.g., rubber[7]), their deficiency in the other property is why researchers continue to look to biology for inspiration. For example, because the major ampullate (dragline) silk of *Caerostris darwini* is stiff ($E_{init} \approx 12$ GPa) and moderately extensible ($\varepsilon_{max} \approx 50\%$)[8], this silk is tougher—that is, it can absorb more energy before failure—than inextensible materials like high tensile steel, aromatic polymer fibers (e.g., Kevlar and M5), and carbon nanotubes[9,10]. The toughness of spider silk, coupled with its low density, is why there has been an intense research focus on unraveling and mimicking its molecular design for lightweight, high strength functional materials[3,11]. However, replicating spider silks is challenging absent the intricate biomachinery spiders use to produce and secrete these supramolecular protein assemblies[12]. More importantly, if our ultimate goal is to develop materials that surpass the performance of natural materials, rather than purely mimic them, we must borrow and combine beneficial and disparate ideas from nature.

The properties of biopolymers are derived from their ability to self-assemble from linear chains of amino acids into complex secondary (e.g., α-helices and β-sheets) and tertiary (e.g., rods and globules) structures stabilized by hydrogen bonds. Highly ordered β-sheet domains are generally found in stiff, fibrillar biopolymers[13–16], whereas amorphous or helical domains are found in extensible biopolymers[2,17]. The combination of these two distinctly different structural domains grants dragline silks their desirable mechanical properties[10,13], but is a design principle that requires the ordered aggregation of multiple polymer chains. Robust biopolymers produced from a single polymer chain would be simpler to fabricate, and exhibit enhanced mechanical properties due to their minimized cross-sectional area.

Inspired by the β-spiral structures found in elastin[18] and flagelliform (capture) silk[19], here we consider design principles to fortify this molecular architecture. Since β-spirals transition between coiled and extended conformations, preventing misfolding is crucial to obtaining and preserving the optimal properties of such structures. We reason that replacing the spiral's helical loops with cyclic peptide rings will constrain the conformation of the repeating unit, reducing the degree of freedom of the biopolymer and the likelihood of misfolding. Monomeric cyclic peptide rings self-assemble into high aspect ratio nanomaterials stabilized by hydrogen bonds between adjacent rings[20]. These assemblies exhibit elastic moduli comparable to dragline spider silks, but their extensibilities are an order of magnitude lower[21]. Using molecular dynamics (MD) simulations, we demonstrate that high extensibility biopolymers can be formed through side-chain-to-side-chain polymerization of cyclic peptide rings, without perturbing self-assembly into rigid nanomaterials. Also, we experimentally verify polymerized cyclic peptides can transition between self-assembled and disordered conformations on demand. MD simulations predict the combination of stiffness and extensibility exhibited by cyclic peptide polymers results in toughness values that exceed 1 GJ m$^{-3}$, whereas dragline spider silks are an order of magnitude lower[8]. Furthermore, this molecular architecture offers precise control over the displayed chemistries; thus, the functionality of these biopolymers can be tailored for applications in a wide range of biological and technological areas.

## Results

**Computational analysis of cyclic peptide assemblies.** Assemblies based on cyclic β-peptides would offer some benefits over their α-amino acid counterparts. Due to the extra methylene group in the peptide backbone, all amide C=O groups are oriented in one direction with respect to the plane of the ring while the N–H groups are oriented in the opposite direction (Fig. 1a). Each ring has a strong dipole moment that drives self-assembly, with the constructive summation of dipole moments yielding increasingly favorable association energies[22]. Since polarization of the assembly increases as rings stack, the length of hydrogen bonds decreases, and their strength increases, as well[23,24].

Another benefit of β-peptides is sequences as short as three amino acids can be cyclized, whereas at least six amino acids are required for D,L-α-peptides. Since the elastic modulus is defined as the bending rigidity ($C_B$) divided by the moment of inertia ($I_0 = \pi d^4/64$ for a cylindrical rod), the cross-sectional area of a cyclic peptide assembly greatly impacts its elastic modulus. Here we perform MD simulations of cyclic β-peptide assemblies (*cyclo* [(-β-Ala)$_n$-]; $n = 3, 4, 6, 8$) under tension to quantify the effect of the ring size on the stiffness of the assembly (Fig. 1a, Supplementary Fig. 1). Our simulations show that the elastic modulus of *cyclo*[(-β-Ala)$_3$-] is double the modulus of *cyclo*[(-β-Ala)$_6$-] (Fig. 1b). We also compute that cyclic β-peptide assemblies have tensile strengths (2–3 GPa) that are a factor of two higher than dragline spider silks[8], but the toughness of the assemblies ($\approx 0.1$ GJ m$^{-3}$) are on par with dragline silks due to the inextensibility of the assemblies ($\varepsilon_{max} \approx 7\%$).

Couet et al. demonstrated that polymer chains can be grafted to cyclic peptide assemblies to form peptide–polymer hybrids[25]. While they focus on the functionality of these hybrids rather than their mechanical properties, we note that this strategy would grant extensibility, but the cross-sectional areas of the hybrids increase and their lengths decrease with increasing polymer chain lengths[26]. To determine how polymer graft length would affect the mechanical properties of cyclic β-tripeptide assemblies, we constructed molecular models of *cyclo*[(-β-HLys-β-Ala-β-HLys)-] with oligoethylene glycol grafts (OEG; (CH$_2$CH$_2$O)$_n$CH$_3$; $n = 5, 15, 25$) attached to all lysines via succinimidyl esters (Fig. 1a). Our computational analysis reveals that the elastic moduli and tensile strengths diminish as a function of OEG length; however, the toughness values are statistically the same as *cyclo*[(-β-Ala)$_3$-] due to their increased extensibility (Fig. 1b; Supplementary Fig. 2). The stiffness, strength, and toughness of cyclic peptide assemblies, like β-sheet domains in spider silks, arise from the confinement and uniform orientation of hydrogen bonds that act cooperatively[14]. If the mechanical properties of the assemblies are to be preserved in a polymeric system, a strategy must be employed that does not impact the ability of the rings to stack, and minimally increases the cross-sectional area.

Previously, we computationally determined that the polymerization of cyclic β-tripeptides via their side chains (e.g., condensation reaction between lysine and glutamic acid) would not disrupt self-assembly as long as the linkage between subunits is at least six atoms in length[22]. This strategy would meet the aforementioned criteria. Here, we compute that increasing the size of the rings from three to eight amino acids negatively impacts stiffness and tensile strength, but greatly enhances extensibility (Fig. 1b; Supplementary Fig. 3a). Alternatively, increasing the linkage length between tripeptide rings from 8 to 12 atoms increases extensibility with less of an impact on stiffness and tensile strength than increasing the ring size (Fig. 1b; Supplementary Fig. 3b). The combination of stiffness and extensibility displayed by our cyclic β-peptide polymers is unmatched by biopolymers found in nature or other high strength synthetic materials (Fig. 1c). We observe the

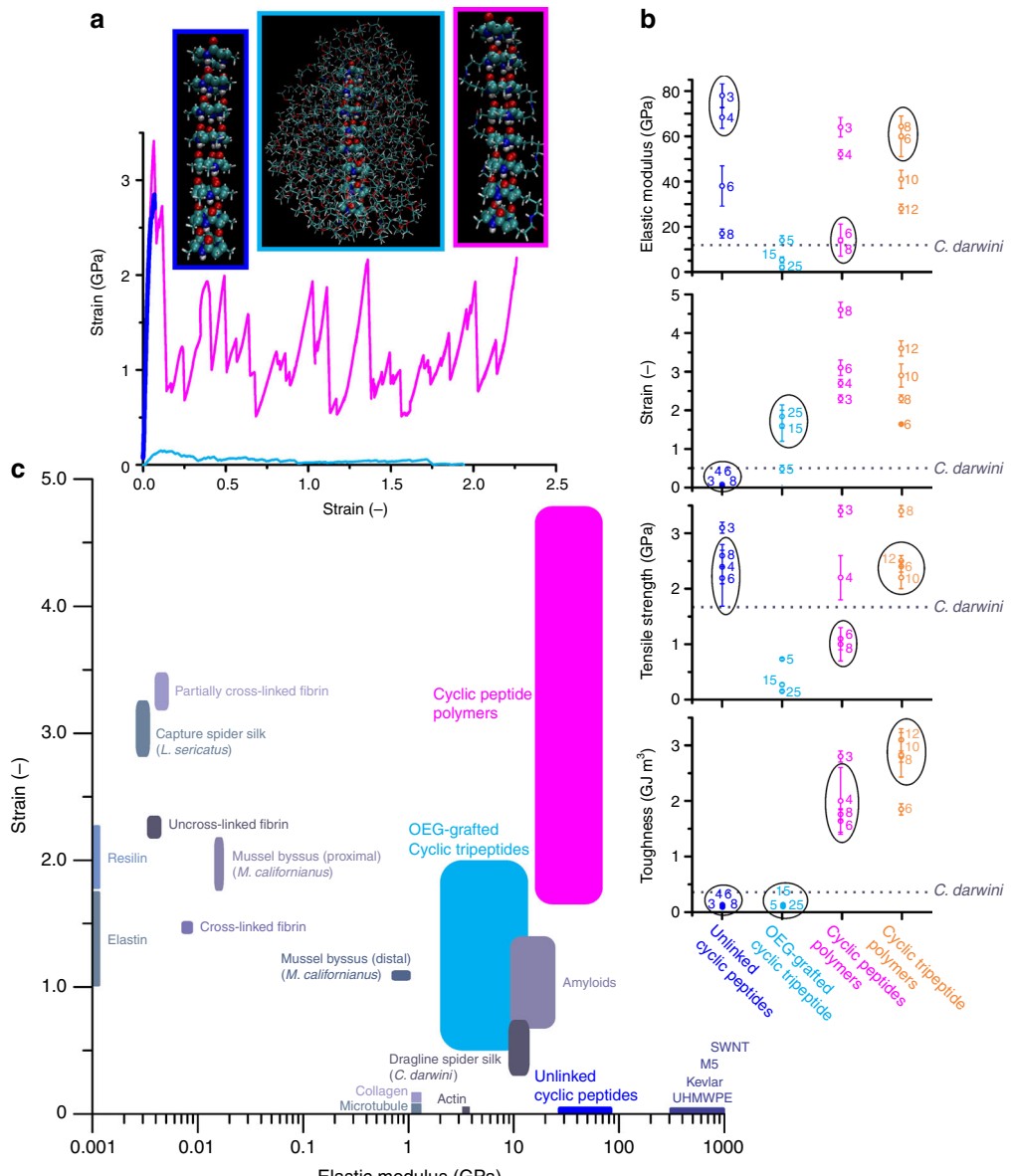

**Fig. 1** Molecular dynamics simulations of cyclic peptide assemblies. **a** Representative stress-strain curves of unlinked (*cyclo*[(-β-Ala)₃-]; blue), OEG-grafted ((CH₂CH₂O)₂₅CH₃; cyan), and polymerized (*cyclo*[(-β-HLys-β-Ala-β-HLys)-] + *cyclo*[(-β-HGlu-β-Ala-β-HGlu)-]; magenta) assemblies consisting of eight subunits. In MD simulations, α-carbon atoms in the backbone of the bottom rings are held fixed, while a spring with a spring constant of 5000 kJ mol⁻¹ is attached to the center of mass of the top ring and pulled upward along the fibril axis at a constant velocity of $5 \times 10^{-5}$ nm ps⁻¹. **b** Computed elastic moduli, strain to failures, tensile strengths, and toughness values for the various cyclic peptide based assemblies (mean ± standard deviation (SD); N = 3). Dotted lines denote the reported values of dragline spider silk from *C. darwini*[8]. Numbers next to data points indicate the number of amino acids in each subunit (blue and magenta), number of OEG repeating units (cyan), or number of atoms in the linkage between subunits (orange). Circles denote data pairings that are not statistically different based on unpaired *t* tests (*p* < 0.05). **c** Computed elastic moduli and strain to failures of cyclic β-peptide assemblies, in comparison with reported values for natural and synthetic materials[4, 8, 9, 17, 34, 54–60]. Single-walled carbon nanotubes are abbreviated as SWNT, and ultra-high molecular weight polyethylene is abbreviated as UHMWPE

continued dissipation of energy at large strains (up to 460%), as is denoted by the saw-toothed features in the simulated stress–strain curves of the cyclic β-peptide polymers (Fig. 1a; Supplementary Fig. 3a, b). After the hydrogen bonds between a pair of rings fail, the assemblies repeatedly relax and stiffen until the hydrogen bonds fail between all rings. This is contrary to the unlinked and OEG-grafted cyclic peptide assemblies where failure of the hydrogen bonds between a single pair of rings leads to the failure of the assembly as a whole, albeit complete failure is delayed in the case of the OEG-grafted assemblies due to the entanglement of the grafts. We note MD simulations cannot determine the failure point of

the contiguous cyclic peptide polymers; therefore, we stopped simulations once all hydrogen bonds were broken and the force reached 500 pN, which is well under the force required to rupture covalent bonds[27]. Even using this conservative treatment, we compute that the toughness values of the cyclic peptide polymers (1.8–3.1 GJ m⁻³) are an order of magnitude higher than the toughest reported spider silk, *C. darwini* (0.35 GJ m⁻³)[8].

**Polymer synthesis and characterization.** We initiated polymer synthesis with a cyclic β-tripeptide system consisting of linkages

eight atoms in length due to this system having the highest predicted stiffness. We selected *cyclo*[β-HGlu-β-HOrn(2-ClZ)-β-HGlu] and *cyclo*[β-HLys-β-HOrn(2-ClZ)-β-HLys] as our sub-units, reasoning that polymerizing a diacid with a diamine would prevent intramolecular lactam formation—a possible side reaction between lysine and glutamic acid residues when present in the same subunit. Ornithine, an amino acid whose amine-terminated side-chain is a methylene group shorter than lysine, was selected as the third residue to impart water solubility for ease in processing. Also, we calculated that electrostatic repulsions between ornithine residues on adjacent rings would inhibit hydrogen bonding between the rings[22], allowing us to induce transitions between rigid and soft conformations without physically manipulating fibers. We undertook the polymerization in a highly concentrated solution of lithium chloride (LiCl) in dimethylformamide (DMF), expecting the Li$^+$ and Cl$^-$ ions to coordinate with the COOH and NH$_2$ groups of the glutamic acid and lysine side chains (Fig. 2a). Under this scenario, interactions between the ions and the side chains prevent excessive interactions between neighboring subunits leading to the formation of two amide bonds between a single pair of subunits. Based on gel permeation chromatography, we estimate that the molecular weight of our resulting polymer is ca. 40 kDa (Supplementary Fig. 4).

When deposited under acidic conditions, we observe agglomerations that were up to 150 nm in height by atomic force microscopy (AFM) (Fig. 3a). X-ray photoelectron spectroscopy (XPS) reveal that the N 1s spectrum of the surface-deposited polymer (Fig. 3b) contains components at binding energies corresponding to neutral (≈ 399.9 eV) and protonated amino groups (≈401.4 eV)[28–31]. The fraction of protonated amino groups ($f_{NH(+)}$) decreases with increasing pH of the deposition solution and all amino groups are deprotonated at pH 11.5 and above. Nanorods are present on mica (Fig. 3d) and graphite (Fig. 3e) when the polymer is deposited under basic conditions, which confirms that self-assembly can be modulated electrostatically. Nanorods are ca. 2 nm in height (Fig. 3d), which is in good agreement with the molecular model (Fig. 3f), and generally

150–300 nm in length with some in excess of 1 μm. Basic polymer solutions that aged over a month, as well as high concentration solutions (≥10 mg mL$^{-1}$) form gels with nanorods ranging in diameter from 2 to 6 nm (Supplementary Figs. 4 and 5). We do not thoroughly investigate the properties of these gels in this report. Rather, we focus on the unique opportunity to evaluate the stiffness of cyclic peptide nanorods, a single subunit in width, for comparison with our computational analysis.

To estimate the persistence length of the nanorods, we processed AFM images using the open-source fiber tracking software FiberApp[32]. We fit the mean-squared end-to-end distances $\langle R^2 \rangle$ of contour segments in 418 individual nanorods to the following worm-like chain model (Fig. 4)[33]:

$$\langle R^2 \rangle = 4\lambda \left[ l - 2\lambda \left( 1 - e^{\frac{-l}{2\lambda}} \right) \right] \quad (1)$$

where $\lambda$ is the persistence length and $R$ is the direct distance between the ends of a contour segment with an arc length of $l$. This model provides a much better fit than the bond correlation function and the mean-squared midpoint displacement model (Supplementary Fig. 6). Based on the estimate of the persistence length, we calculate the bending rigidity ($C_B$) of the nanorods to be $4.0 \times 10^{-26} \pm 5 \times 10^{-27}$ N m$^2$ (mean ± SD).

Defining the cross-sectional geometry of the nanorods is somewhat ambiguous because it relies on the position of amino acid side chains whose lengths are of the same order as the cyclic peptide backbone. Therefore, we conservatively assume the nanorods have a solid, circular cross-section. Using the average AFM height ($d = 2.0$ nm), we calculate the moment of inertia ($I_0 = \pi d^4/64$) to be $7.85 \times 10^{-37}$ m$^4$, which yields an elastic modulus ($E = C_B/I_0$) of 51 ± 6 GPa (mean ± SD) for the nanorods. Even by this conservative estimate, the elastic modulus of our nanorods is amongst the highest reported values for amino acid-based assemblies[8,16,21,34–36]. Moreover, AFM experiments demonstrate that the rigidity of the polymer can be altered on command, which means mechanically deformed nanorods could be similarly treated to restore their folded conformation.

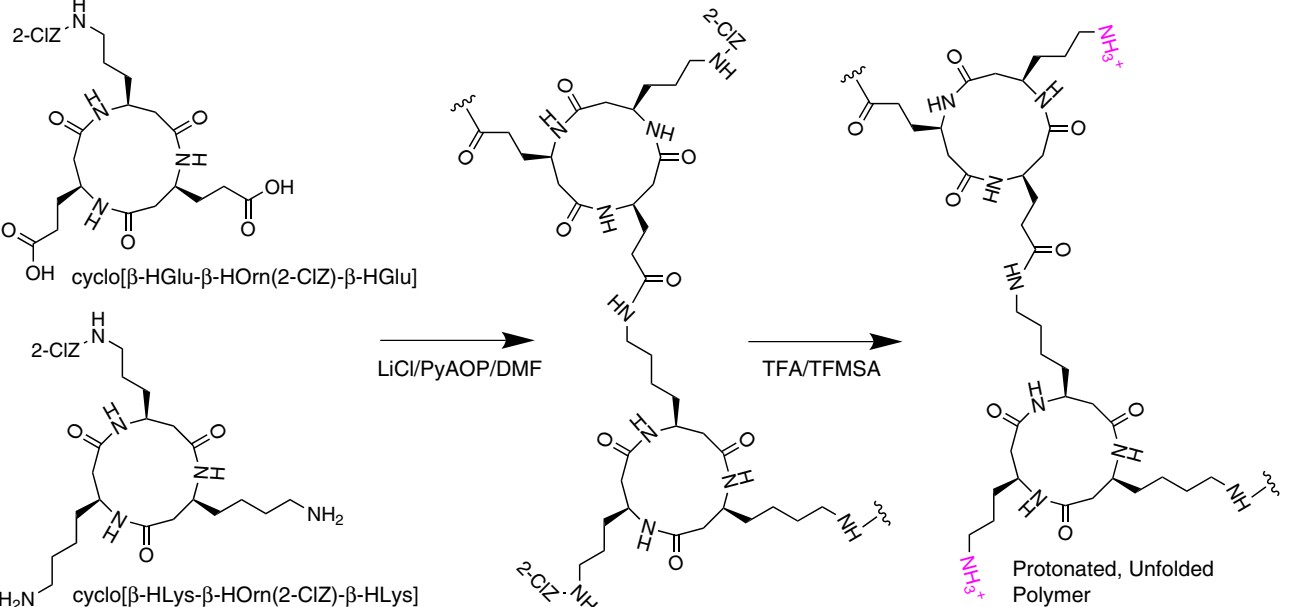

**Fig. 2** Cyclic β-tripeptide polymer synthesis scheme. Cyclic peptides were polymerized using the coupling agent (7-azabenzotriazol-1-yloxy) tripyrrolidinophosphonium hexafluorophosphate (PyAOP) and final deprotection was performed with a mixture comprising trifluoroacetic acid (TFA) and trifluoromethane sulfonic acid (TFMSA). Synthesis details are in the Methods section and Supplementary information

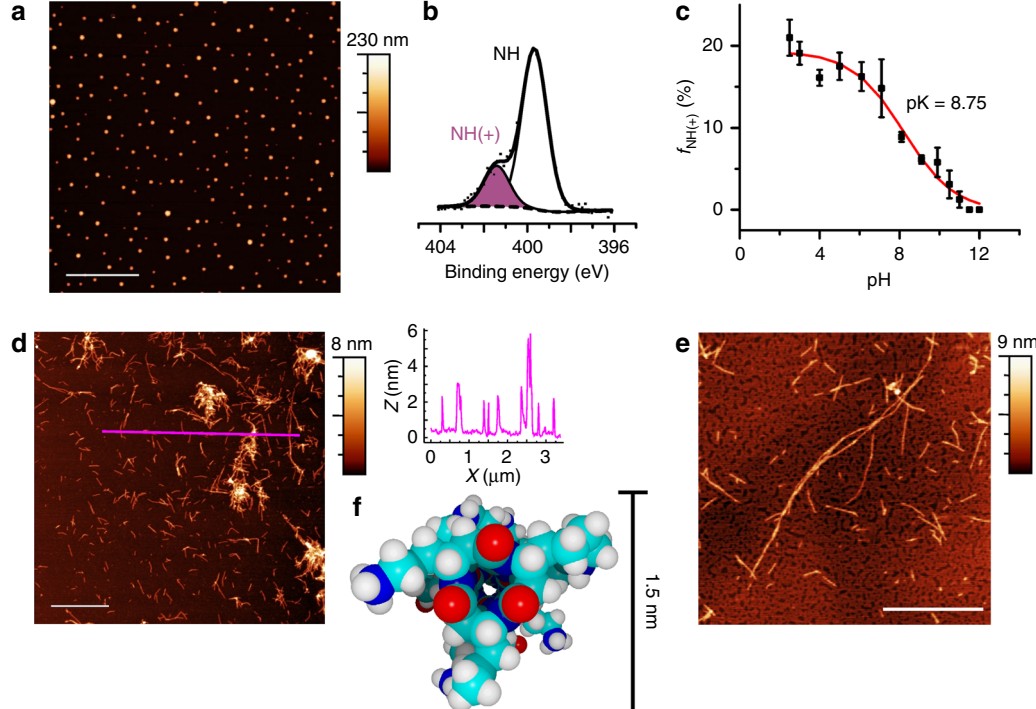

**Fig. 3** pH response of the cyclic β-tripeptide polymer. **a** AFM image (5 μm scale bar) of polymer deposited on mica from a 1 mg mL⁻¹ solution in 0.1 M acetic acid (pH = 2.9). Color bar indicates the Z heights in the images, with the baseline (black) set to 0 nm. **b** N 1s XPS spectrum collected from the polymer-coated substrate shown in (**a**). **c** Fraction of protonated amino groups ($f_{NH(+)}$), as determined by XPS, in thick polymer films deposited on gold from 0.1 M sodium phosphate buffers with pHs ranging from of 2.5–12.0. Data points (mean ± SD; $N = 3$) are fit to a sigmoid curve (see Methods section)[28], to determine the pK. AFM images of the polymer deposited on (**d**) mica (1 μm scale bar) and (**e**) graphite (500 nm scale bar) from a 1.0 mg mL⁻¹ aqueous solution after NH₄OH vapor diffusion (pH ≈ 11); height profile along the magenta line is shown to the right (**d**). **f** Cross-sectional view of a molecular model of the synthesized cyclic β-tripeptide polymer

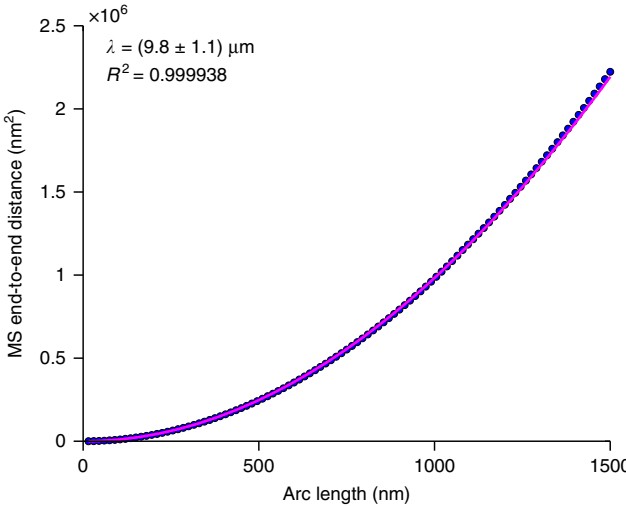

**Fig. 4** Persistence length of nanorods. Mean-squared (MS) end-to-end distance of nanorods as a function of their inner contour length in 100 bins (blue dots). Data was collected from AFM images of nanorods deposited on mica and graphite from aqueous solutions after NH₄OH vapor diffusion. The magenta line represents the fit of the data to Eq. (1), which is used to determine the persistence length (λ) (mean ± SD; $N = 418$)

In addition to modulating self-assembly, the displayed primary amines allow for the use of these nanorods as molecular scaffolds. To demonstrate this functionality, we exposed the polymer to 1.4 nm Au nanoparticles, each bearing a single amine-reactive succinimidyl ester. After functionalization, nanorods appear to have increased in height from ca. 2 to ca. 6 nm (Fig. 5a, b), indicating that nanoparticles are attached to nanorods. XPS also corroborates the presence of Au (Fig. 5c). Furthermore, we observe a distinct, ribbed topography along the long axis of decorated nanorods, which is consistent with our model showing a helical display of ornithine side chains (Fig. 5d).

## Discussion

We have presented an orthogonal strategy—that is, each reaction step proceeded without interfering with a subsequent chemical reaction—for synthesizing cyclic peptide subunits for copolymerization via side-chain-to-side-chain amide linkage formation. Our initial design resulted in the fabrication of a biopolymer that self-assembles into nanorods composed of a single structural domain that is both stiff and highly extensible. Due to the compact nature of the structural domain in our cyclic β-peptide polymers, which are stabilized by covalent and hydrogen bonds, computational analysis yielded tensile strengths greater than high performance polymer fibers (e.g., M5 and Kevlar) and toughness values that far exceed other natural or synthetic materials. We emphasize that fibers consisting of a single polymer strand display these mechanical properties. This not only means that these polymers can be used as structural and functional materials at the nanoscale, but also as hierarchical assemblies that will likely exhibit enhanced mechanical properties akin to protein assemblies found in nature[2,5]. While we were able to obtain polymer gels without cross-linkers, cross-linking could be used to promote and stabilize the formation of higher order assemblies, and further manipulate their mechanical properties. Liu *et al.* reported a

50% increase in the elastic limit and extensibility of partially cross-linked fibrin fibers versus uncross-linked fibers, whereas, fully cross-linked fibers were twice as stiff but 35% less extensible (Fig. 5d)[17,37]. This highlights that, as impressive as the

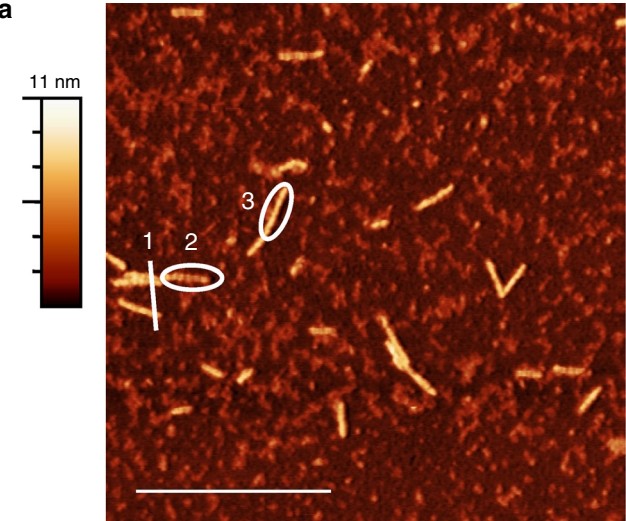

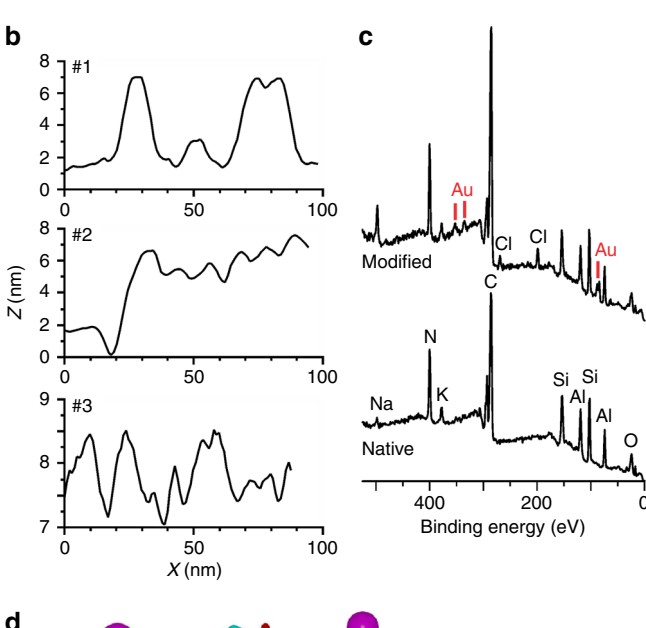

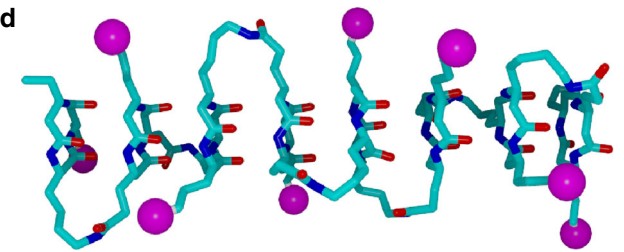

**Fig. 5** Nanorod functionalization. **a** AFM image (300 nm scale bar) of nanorods functionalized with 1.4 nm Au nanoparticles deposited on mica from a basic solution (0.1 mg mL$^{-1}$; pH ≈ 11). Color bar indicates the $Z$ heights in the images, with the baseline (black) set to 0 nm. **b** Height profiles are shown across (1) and along the long axis (2 and 3) of decorated nanorods. **c** XPS survey scan confirming the presence of Au after functionalization. **d** Molecular model of the cyclic β-tripeptide polymer. Amine groups on ornithine residues are represented as magenta balls to highlight their helical display

mechanical properties we have reported are, we have only begun to explore the potential of these assemblies.

Moving beyond structural applications, cyclic peptide assemblies have been successfully used in biotechnological, optical, electronic, and chemical applications[38–41]. The elegant spatial display of functionalizable side chains positions these assemblies as molecular scaffolds for nanoscale assembly. While here we chose ornithine as the third residue, our strategy is easily adaptable to residues bearing a wide range of chemical functionality, such as carboxylic acids, alternating amino groups and carboxylic acids, fluorophores, and π-stacking moieties. Larger cyclic peptide rings could also incorporate more than one displayed functional group to create multifunctional nanostructures. In addition to utilizing displayed side-chain chemistries, the polarization of the hydrogen bonds in the assemblies could potentially be exploited for nanopiezotronics[42,43], or spintronics[44]. In summary, we have presented a versatile system that can meet the structural and functional demands of emerging nanotechnology driven devices.

## Methods

**Synthesis of cyclic β-tripeptide subunits.** Fmoc-β-HGlu(O*t*-Bu)-OH and Fmoc-β-HLys(Boc)-OH were obtained from Biosciences. Fmoc-β-HOrn(2-ClZ)-OH was synthesized via the Arndt-Eistert homologation[45,46]. The linear precursors to *cyclo* [β-HGlu-β-HOrn(2-ClZ)-β-HGlu] and *cyclo*[β-HLys-β-HOrn(2-ClZ)-β-HLys] were synthesized using the highly acid-labile 2-chlorotrityl chloride resin[47]. Cyclization of the peptides with 1-[bis(dimethylamino)methylene]-1*H*-1,2,3-triazolo[4,5-*b*]pyridinium 3-oxid hexafluorophosphate in DMF, purification by gel permeation chromatography, and side-chain deprotection with trifluoroacetic acid (TFA)/water/triisopropylsilane furnished the cyclic β-tripeptide subunits, both in roughly 90% overall yield (detailed procedures provided in Supplementary Note 1).

**Polymerization of cyclic β-tripeptide subunits.** In total, 36 mg of each subunit along with 130 mg of (7-azabenzotriazol-1-yloxy)tripyrrolidinophosphonium hexafluorophosphate (PyAOP) (0.248 mmol, 4 eq.) was dissolved in 1 mL of 2 M LiCl in DMF under argon. To this mixture, 0.11 mL of *N,N*-diisopropylethylamine (DIEA) was added when a slight darkening of the solution was observed. The mixture was then stirred under argon for 68 h. Subsequently, DMF was removed from the mixture when a sticky product resulted. This product was treated with 18 MΩ–cm deionized (DI) water and, upon sonication, the product was dislodged from the flask wall. The suspension of the product in DI water was centrifuged in a 2 mL tube and washed consecutively with water and $CH_3CN$. The washed product was dried initially using a vacuum pump then lyophilized for 48 h, yielding 45 mg of cyclic β-tripeptide polymer.

**Deprotection of 2-Cl-Z group.** Lyophilized polymer (25 mg) was taken in a round bottom flask with a stir bar, and 75 mL of thioanisole was added under argon. The mixture was cooled in an ice bath. TFA (0.5 mL) was added to the mixture followed by 10 min of stirring. Then, 50 mL of trifluoromethane sulfonic acid was added dropwise under vigorous stirring at 0 °C, to efficiently contain/disperse the generated heat. The mixture was taken out of the ice bath after 15 min and the reaction was allowed to proceed further for 2 h. Cold diethyl ether (Et$_2$O) was added to the flask to the brim and placed in a freezer to facilitate the precipitation of the product. Et$_2$O was then siphoned off and the procedure was repeated a dozen times. The final precipitate after washing was dried in an oil bath at 33 °C. The precipitate was further dried under vacuum for 12 h, yielding 22 mg of the deprotected polymer.

**Anion exchange of the deprotected polymer.** AG MP-1 anion exchange resin (0.6 g) (Bio-Rad) (selectivity: OH$^-$ = 1.0; Cl$^-$ = 22; and bed volume = ~1.3 mL) was treated with 29 mL of 1 M NaOH in a conical flask (i.e., selectivity difference × bed volume = 22 × ~1.3 = ~29 mL of 1 M NaOH is necessary to exchange Cl$^-$ ions with OH$^-$ ions). The flask was purged with argon and stoppered overnight to prevent absorption of CO$_2$ and conversion of OH$^-$ to HCO$_3^-$. After washing with DI water, the treated resin was added to a flask containing deprotected polymer in 1 M NaOH, and the flask was agitated gently on a shaker for 1 h. The polymer solution was filtered and lyophilized to collect the ion-exchanged polymer.

**Functionalization with gold nanoparticles.** A polymer solution (0.15 mg mL$^{-1}$) was prepared in phosphate buffered saline (pH 7.4), and a 30 μM solution of 1.4 nm Mono-Sulfo-NHS-Nanogold® labeling reagent (Nanoprobes), capped with a single *N*-hydroxysulfosuccinimide (NHS) ester group, was prepared in DI water. A mixture of the gold solution (0.2 mL) and polymer solution (0.4 mL) was allowed to react for 1 h at room temperature and was then subjected to gel permeation chromatography to isolate gold functionalized polymer.

**Gel permeation chromatography**. Analytical size exclusion chromatography (Prostar HPLC System, Varian) was carried out using a $4.6 \times 250$ mm gel column (Aquagel-OH MIXED M, 8 μm, Agilent) operating isocratically with a DI $H_2O$ mobile phase, adjusted to pH 4 with trifluoroacetic acid, running at 1 mL min$^{-1}$. First, 20 μL of dilute protein standard spanning a 1,350-670,000 Da range (thyroglobulin, γ-globulin, ovalbumin, myoglobin, and vitamin b12, Gel Filtration Standard, Bio-Rad, 151–1901) was injected and chromatograms recorded using dual UV absorbance (Prostar 345, Varian) at 280 and 215 nm. Separately, cyclic peptide–polymer samples were injected and compared to the elution profile of the protein standard. To verify the peak containing polymers, fractions were collected and deposited on mica for AFM and XPS analysis.

**AFM**. Dimensions of polymer nanostructures were measured by AFM in intermittent-contact mode using a Bruker Dimension 6100 (Santa Barbara, CA) instrument. Images were collected on freshly cleaved mica and graphite surfaces. Upon cleavage, surfaces were exposed to polymer solutions for 10 min and then rinsed with DI $H_2O$ to remove any salt formations. Imaging was performed using rectangular probes with 5–10 nm wide tip apex (OTESPA-R3, 26 N/m, 300 kHz, Bruker, Camarillo, CA) at a scan rate of 0.5 Hz. Images ranged from 512 to 1024 pixels in both fast and slow axis. End- and mid-points for nanorod length calculations were collected using Gwyddion v2.37 (Czech Metrology Institute, Czech Republic) software. Nanorods selected for rigidity analysis were clearly not in contact any other nanorods, and were longer than 50 nm.

**XPS**. The dissociation constant of the polymer was determined by in vacuo XPS on a spectrometer equipped with a monochromated Al Kα X-ray source operating at $1486.6 \pm 0.2$ eV and a flood gun emitting low energy electrons (≤10 eV) and Ar$^+$ ions for charge neutralization. Polymer solutions (1.0 mg mL$^{-1}$) were prepared in 100 mM sodium phosphate (pH 2.5–12.0); pH was adjusted using NaOH and HCl. Thick films were cast on gold-coated silicon (ca. 100 nm Au) wafers (Platypus Technologies) by pipetting 2.0 μL drops, three drops per pH, onto the substrates and allowing the drops to dry at room temperature. Prior to polymer deposition, substrates were cleaned by sequential sonication for 5 min in 0.005% (v/v) Triton® X-100, piranha wash [7:3 $H_2SO_4$ (98 wt%)/$H_2O_2$ (30 wt%)], and RCA standard clean 1 [1:1:5 $NH_4OH$ (28.0–30.0% $NH_3$ basis)/$H_2O_2$ (30 wt%)/$H_2O$]. (Caution: Piranha solution is extremely oxidizing, reacts violently with organics, and should only be stored in loosely covered containers to avoid pressure buildup.) Substrates were rinsed with 18 MΩ–cm $H_2O$ after each cleaning step and dried under a stream of nitrogen gas at the completion of the cleaning process.

Survey and high-resolution spectra were collected at pass energies of 200 eV (1.0 eV step size) and 20 eV (0.10 eV step size), respectively. Spectra were deconvoluted in Unifit (ver. 2011), using a combination of Lorentzian and Gaussian line shapes for the individual components and Shirley and linear functions for the baselines[48]. Contributions to the full-width at half-maximum from Lorentzian line shapes were fixed at 0.1 eV, and Gaussian line shapes were constrained between 1.2 and 1.3 eV. Elemental compositions were quantified using calibrated analyzer transmission functions, Scofield sensitivity factors[49], and effective attenuation lengths for photoelectrons that were calculated using the standard TPP-2M formalism[50,51].

The dissociation constant (pK) of the polymer was determined by fitting the fraction of protonated amines ($f_{NH(+)}$), as a function of pH, to the following equation and minimizing the sum of the squared errors:

$$f_{NH(+)} = A\left[\frac{10^{B(pK-pH)}}{1 + 10^{B(pK-pH)}}\right] + C \qquad (2)$$

where $A$ is the amplitude, $B$ is related slope of the transition region, $C$ is the minimum value, and pK is the pH at half-maximum.

**Computational analysis**. Molecular models were initially generated using Hyperchem (Hypercube, Inc.) and optimized using the CHARMM27 molecular mechanics force field under the steepest descent algorithm with a terminal condition of 0.01 kcal Å$^{-1}$ mol$^{-1}$ or 1000 cycles. MD simulations were conducted in vacuum using GROMACS (ver. 2016.4) the CHARMM36m protein force field[52]. All models underwent the same set of protocols through a leap-frog time integration with a 0.002 ps time step: (1) energy minimization (no time marching during this step), (2) a constant volume and temperature (NVT) ensemble, with all heavy atoms constrained, performed over 1 ns with the temperature controlled at 298 K by rescaling the molecular velocities every 1 ps, (3) a NVT ensemble, with all $C_\alpha$ atoms constrained, performed over 1 ns with the temperature controlled at 298 K by rescaling the molecular velocities every 1 ps, (4) a constant volume and constant energy (NVE) ensemble, with all $C_\alpha$ atoms constrained, performed over 5 ns, and (5) a NVE ensemble, with all atoms allowed to move freely, performed over 5 ns. To simulate single molecular tensile tests, $C_\alpha$ atoms on the bottom cyclic β-peptide subunit were fixed throughout the simulation and a spring with a spring constant of 5000 kJ mol$^{-1}$ was attached to the center of mass of the top subunit and pulled upward along the fibril axis at a constant velocity of $5 \times 10^{-5}$ nm ps$^{-1}$. Mechanical properties reported represent the mean value and 95% CI calculated by averaging the results of three simulations, with an additional NVE simulation using the previous starting structure performed over 1 ns to sample different

conformations. In the NVT and NVE ensembles, the length of the bonds between hydrogen and heavy atoms were restrained by using the SHAKE method[53].

## Data availability

Data supporting the findings of this study are available from the corresponding author upon reasonable request.

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

## Acknowledgments

This work was supported by the Office of Naval Research through the basic research program and Institute for Nanoscience at the Naval Research Laboratory. X.L. and R.A.L. were supported by "RESBIO—The National Resource for Polymeric Biomaterials" funded under NIH Grant No. P41 EB001046-01A1 and the Center for Advanced Fibers and Films (CAEFF) at Clemson University. Computational support was provided by the Palmetto High Performance Computing Resource at Clemson University. Also, the authors thank Dr. Dmitri Y. Petrovykh for his help initiating the project and fruitful discussions.

## Author contributions

K.P.F. conceived and executed computational experiments, performed computational, and experimental data analysis, and wrote the paper. N.B. and X.L. assisted with computational experiments and analysis. X.L. was supervised by R.A.L. M.K.K.-V., and T.D.C conceived the polymer synthesis scheme. M.K.K.-V. performed polymer synthesis under the supervision of T.D.C., K.P.F, M.K.K.-V., D.E.B, C.R.S., K.J.W., and J.L.K. performed material characterization under the supervision of T.D.C.

## Additional information

**Competing interests:** The authors declare no competing interests.

