## [Peer Review File · Nature Communications]

REVIEWERS' COMMENTS:

Reviewer #1 (Remarks to the Author):

I am satisfied that the authors have mostly addressed the items from my previous review.

There are a few items that need to be addressed prior to my acceptance:

1. In the abstract, the phrase "incredibly high" is sensationalized and unscientific language. Can the authors please rephrase to give this appropriate quantifiable context, ie "toughness greater than or comparable to X" or similar.
2. Strain is a dimensionless quantity, so why are the e.g. axis labels for strain given as nm/nm?? This seems really weird. What is the point the authors are trying to make? The authors should state this in the manuscript.
3. Page 10, "balls" should be spherical shapes or similar. Again a problem with unscientific language.

Reviewer #4 (Remarks to the Author):

I enjoyed reading the revised paper and the responses. This is an extremely interesting piece of work that will make an important contribution to the literature. The manuscript is in good shape and I recommend publication as is. I have examined the answers and list of changes to address the earlier concerns and found them, to the extent I can judge, adequate. They have markedly improved the manuscript.

Response to Reviewers

REVIEWERS' COMMENTS:

Reviewer #1 (Remarks to the Author):

I am satisfied that the authors have mostly addressed the items from my previous review.

There are a few items that need to be addressed prior to my acceptance:

1. In the abstract, the phrase "incredibly high" is sensationalized and unscientific language. Can the authors please rephrase to give this appropriate quantifiable context, ie "toughness greater than or comparable to X" or similar.

The sentence has been rephrased in the revised manuscript.

2. Strain is a dimensionless quantity, so why are the e.g. axis labels for strain given as nm/nm?? This seems really weird. What is the point the authors are trying to make? The authors should state this in the manuscript.

The axis labels have been changed from (nm/nm) to (-) to indicate that the units of the axis are dimensionless.

3. Page 10, "balls" should be spherical shapes or similar. Again a problem with unscientific language.

This word has been replaced by agglomerates in the revised manuscript.

Reviewer #4 (Remarks to the Author):

I enjoyed reading the revised paper and the responses. This is an extremely interesting piece of work that will make an important contribution to the literature. The manuscript is in good shape and I recommend publication as is. I have examined the answers and list of changes to address the earlier concerns and found them, to the extent I can judge, adequate. They have markedly improved the manuscript.